# A Combined Catalytic Ozonation-MBR Approach to Remove Contaminants from the Mature Landfill Leachate in the Yellow River Basin

**DOI:** 10.3390/toxics10090505

**Published:** 2022-08-28

**Authors:** Cui Ma, Panfeng Ma, Zhengguang He, Xiao Mi

**Affiliations:** 1School of Environmental and Municipal Engineering, North China University of Water Resources and Electric Power, Zhengzhou 450046, China; 2School of Ecology and Environment, Zhengzhou University, Zhengzhou 450001, China

**Keywords:** mature landfill leachate, catalytic ozonation, membrane bioreactor, metagenomic approach, integrated process

## Abstract

The mature landfill leachate (MLL) is characterized by a large number of fulvic acids and humic acids, which is refractory organic matter and can be cleaned by ozone oxidation. However, the poor property of mass transfer prohibits the widespread use of ozone oxidation in actual leachate treatment. Meanwhile, some combined processes are adopted to treat the mature landfill leachate, which places catalytic ozonation before the membrane bioreactor (MBR) process to enhance the biodegradability of MLL. Thus, this research is conducted to investigate the practicability of applying nano-Fe_3_O_4_ loaded cow-dung ash (Fe_3_O_4_@CDA) and biological post-treatment with MBR for the effective removal of pollutants from MLL and puts forward the variation of organics in leachate between catalytic ozonation and MBR. The addition of catalytic ozonation not only improved the removal of hazardous organics but also enhanced the biodegradability of the leachate and favored the subsequent MBR process. Chemical oxygen demand (COD) removal in the catalytic ozonation step was optimized, and 53% removal was obtained at pH = 7, catalyst dosage = 1.0 g/L, and O_3_ dosage = 3.0 g/L. After the MBR process, COD in effluent stabilized in the range of 57.85–65.38 mg/L, and the variation range of the ammonia nitrogen (NH_3_-N) concentration was 5.98–10.24 mg/L. The catalytic ozonation-MBR integrated process showed strong feasibility in dealing with the biologically pre-treated leachate.

## 1. Introduction

In 2019, the generation of municipal solid waste (MSW) rapidly expanded, and the amount of MSW is about 2.35 hundred million tons in China’s large and medium-sized cities [1]. Sanitary landfilling has been the most commonly used method for treating municipal solid waste. Generally, leachate is generated due to the interaction between waste and precipitation during landfilling. The landfill leachate is characterized by high concentrations of toxic organics (carcinogen, mutagen), nitrogen oxides, and inorganic substances [2]. Classically, landfill leachate poses a threat to the surrounding environment and ecosystems. Consequently, more and more attention has been paid to the studies of the effective disposal of landfill leachate by new technologies.

Many methodologies have been applied to treat landfill leachate. Biological treatment is one of the most conventional treatments because of its cost-effectiveness and low effect on the environment. Yet numerous refractory organics still stay in the leachate. Beyond that, poor effects on activated sludge caused by hazardous organic contaminants and heavy metals have been confirmed clearly [3]. Therefore, the type and composition of landfill leachate affect the effectiveness of biological treatment.

Meanwhile, biological treatments are often chosen to deal with young landfill leachate in the short term. Conventional biological treatment has been confirmed to gain a Chemical oxygen demand (COD) removal of 52.5% from young landfill leachate with high biodegradability [4]. However, the conventional biological treatment may produce plenty of sewage sludge. Strict discharge standards of leachate usually cannot be sufficed through single biological treatments due to their weaknesses in dealing with the refractory organics. In order to mitigate this limitation, physicochemical methods have been added to the whole treatment process. Integrating the coagulation-flocculation, MBR and reverse osmosis show high effectiveness in dealing with landfill leachate, bringing about a significant removal of COD, ammonia nitrogen (NH_3_-N), and chroma of nearly 85, 90, and 100%, respectively [5]. Especially the MBR process is proved to be one of the more effective methods compared with other conventional treatment technologies, dependent on activated sludge [6], due to its integration of the activated sludge process and the membrane separation. Recently, MBR technology has been one of the most efficient leachate treatment methods. MBR has been reported to perform better with the landfill leachate treatment, including higher effluent quality, more vital process stability, easier management, and less sludge production [7]. Previous reports show that the MBR process has a higher loading rate and COD removal efficiency at a shorter hydraulic retention time [8]. However, it still has ineffectiveness in dealing with some refractory organics.

To eliminate these refractory organics, advanced oxidation processes are often required. Recently, many advanced oxidation processes (AOPs) have played a critical point in the treatment of landfill leachate, such as ozone and ozone-based (O_3_, O_3_/UV/H_2_O_2_), photocatalytic (UV/TiO_2_), Fenton, and photo Fenton (Fe^2+^/H_2_O_2_, Fe^2+^/H_2_O_2_/UV), electrochemical oxidation, etc. [1,9,10,11,12,13]. As one of the methods mentioned above, catalytic ozonation technology performs well in water treatment. Catalytic ozonation can be applied to treat various wastewaters, and the effluent quality is guaranteed [14]. The treatment of raw leachate by catalytic ozonation may be complex due to the high pollutant concentration and color of wastewater and the high operating cost of catalytic ozonation. However, catalytic ozonation may be a good choice for the effluent of biological treatment systems and influent follow-up treatment [15]. Considering the characteristics mentioned above, the catalytic ozonation-MBR system for mature leachate treatment was designed for the scale of a 1000 m^3^/d landfill leachate treatment plant in Zhengzhou.

In this study, the treatment efficiency and mechanism of the catalytic ozonation-MBR process for the secondary biological effluent of landfill leachate were systematically investigated. Furthermore, the molecular weight changes of dissolved organics in treated wastewater using a molecular weight distribution technology were analyzed. The molecular structure of humus and aromaticity of refractory organics in different treatment processes by three-dimensional (3-D) fluorescence spectroscopy was studied throughout the whole system. In the end, the bacterial community composition, which plays a vital role in the degradation of refractory organics in the MBR process, was identified.

## 2. Experimental Section

### 2.1. Leachate Samples

The leachate samples were collected from a landfill in Yuanyang County in Xinxiang (belonging to the Yellow River Basin) after biological treatment. The original treatment process of the landfill was shown in Figure 1. Characteristics of the collected leachate: pH 8.48, biochemical oxygen demand (BOD)/COD 0.05, and COD 1050 mg/L. A sampling of the leachate was carried out during summer and autumn in 2020. All samples were stored at a temperature of 4 ℃ for this work.

### 2.2. Preparation of the Catalyst

The Fe_3_O_4_ nanoparticles loaded cow dung ash (nano-Fe_3_O_4_@CDA) was synthesized through in situ precipitation. 2.5 g cow dung ash was added into a 500 mL flask containing 250 mL distilled water under stirring conditions. The slurry was kept bubbling with N_2_ flow for 30 min to remove oxygen from the solution and then put in a 95 °C water bath. Then, 9.27 g FeSO_4_·7H_2_O was added to the flask. Afterward, 2.7 g NaOH and 2.7 g NaNO_3_ were dissolved in 100 mL distilled water and added slowly (5 mL min^−1^) into the heating solution while stirring violently and stable bubbling with N_2_ flow during the entire process. After that, the solution was kept heating at 95 °C for another hour, then cooled down to room temperature. The deposit was separated by a permanent magnet. The solid was then washed with deionized water and ethanol repeatedly under ultra-sonication. Finally, the formed product (nano-Fe_3_O_4_@CDA) was dried in a vacuum oven at 60 °C for 12 h. Fe_3_O_4_ was prepared as above procedure without adding cow dung ash. All the products were stored in desiccator under ambient temperature for further experiments.

### 2.3. Ozonation Procedure

The catalytic ozonation process was conducted in the reactor with a diameter of 60 mm. The used O_3_ was produced through a 5S-B laboratory O_3_ generator (Zoutai, China). During the reaction, the O_3_ generator provided a continuous supply of O_3_. The ozone micro-bubble sizes were controlled by the aeration devices with a hole size of 0.22 μm). The O_3_ dosage was adjusted by changing the electric voltage. The rest of the O_3_ in the off-gas reacted with a 25% potassium iodide solution.

### 2.4. MBR Process

The MBR reactor was made of a rectangular reactor of perspex material with an effective volume of 2 L. The hollow fiber membrane was used with an aperture of 0.4 μm and an effective area of 0.11 m^2^. The activated sludge used in the experiment was taken from the aerobic tank of the landfill, and the concentration of the sludge injected in the MBR reactor was about 10,000 mg/L. The efficiency of MBR for the effluent after catalytic ozonation under different hydraulic retention times (HRT) (10 h, 16 h, 22 h and 28 h) was investigated.

### 2.5. Analytical Methods

The forms of the Fe_3_O_4_ loaded on the catalyst composite were characterized by XRD. The powder XRD of the catalyst was carried out with Bruker D8 Advance Diffractometer using K_α1_ (λ = 0.1541 nm) with operating voltage of 40 kV and 30 mA. The morphology of the catalyst was characterized by Nova 400 Nano SEM (FEI Company, Hillsboro, OR, USA).

Analyses of chemical oxygen demand (COD), ammonia nitrogen (N-NH_4_), mixed liquor suspended solids (MLSS) and chloride (Cl^−^) were performed based on the standard Chinese NEPA methods [16].

Specific oxygen uptake rate (SOUR) and Extracellular Polymeric Substances (EPS) can effectively reflect the activity of sludge microorganisms, which is one of the important indicators of microbial metabolism. SOUR is analyzed according to the previous literature [7].

Molecular weight (MW) distribution in the leachate was determined by gel permeation chromatography (SHIMADZU, LC-10ADVP). MW is analyzed according to previous literature [17].

Samples collected from the fouled membrane were disposed of in a vacuum freeze drier for 12 h. Next, the Fourier Transform Infrared Spectroscopy (FTIR) of the samples was directly measured by FTIR spectroscopy (Nicolet 6700, Thermo Fisher Scientific Company, Waltham, MA, USA). The FTIR spectra were analyzed based on the Sadtler handbook of Infrared Spectra. 2.4.3 [18].

The concentration of organics in the leachate was measured by F-7100 3D-EEM spectroscopy (Hitachi Company, Tokyo, Japan) with a 150 W xenon lamp. The emission scans were conducted from 250 to 500 nm at 5 nm intervals. The excitation wavelengths were scanned from 220 to 450 nm with 5 nm steps. The scanning speed was held at 1000 nm/min. The samples were filtered through GF/F glass fiber filters and diluted before being detected.

## 3. Results and Discussion

### 3.1. Characterization of the Catalyst

Figure 2 displays the XRD patterns of the nano-Fe_3_O_4_, CDA, and Fe_3_O_4_ nanoparticles loaded with cow-dung ash composites (nano-Fe_3_O_4_@CDA). As shown in Figure 1a, the diffraction peaks at 2θ = 18, 30, 35.5, 37, 43, 53.4, 57, and 62.5° were attributed to the synthesized nano-Fe_3_O_4_, indicating that Fe_3_O_4_ with the high purity was successfully synthesized. 

Clearly, the nano-Fe_3_O_4_ particles presented some irregular appearances, circular or square. Because of the agglomeration phenomenon that could be assigned to the magnetism of nano-Fe_3_O_4_, the nano-Fe_3_O_4_ diameter reckoned using proportions in the SEM micrographs was 100 nm. This was obviously larger than the result calculated using XRD. Figure 3c shows that the roughness was increased after binding of nano-Fe_3_O_4_ on the surface of CDA. Meanwhile, the nano-Fe_3_O_4_ nanoparticles growing on the CDA surface showed better dispersing and less co-aggregation.

### 3.2. Optimization of the Ozonation Process

As shown in Figure 4a, the beneficial effect of increased catalyst dosage on growing COD removal during the catalytic micro-ozonation process was clear. A higher catalyst dosage means more active sites, increasing COD removal. In these experiments, ozone concentration and pH keep constant. As expected, when the catalyst dosage increased, higher COD removal was reached. As shown in Figure 4a, the introduction of Fe_3_O_4_@CDA increases the treatment efficiency of the leachate to some extent. Significantly, Fe_3_O_4_@CDA dosage rose from 0 to 1.0 g/L, giving rise to the enhanced COD removal from 15.9% to 47.8%. No obvious change in COD removal was observed when an increased catalyst dosage was used. In fact, the further increment in the catalyst dosage reduced the treatment performance in an almost imperceptible way. Indeed, similar behaviors were found in the previous literature [12,19] that an excessive catalyst leads to a radical scavenger effect, reducing the amount of HO· available for the catalytic-ozonation of the effluents.

Figure 4b presented the COD removal in the leachate under various O_3_ dosage conditions. The COD removal efficiency in the leachate increased from 23% to 53% when the O_3_ dosage increased from 1.0 to 3.0 g/L. Obtained results suggested that the COD removal efficiency was sensitive to applied O_3_ dosage. However, the COD removal efficiency leveled off at the input O_3_ dosage larger than 3.0 g/L. The active sites of Fe_3_O_4_@CDA were limited when the catalysts were quantitative [1].

#### 3.2.1. Spectroscopic Analysis of the Dissolved Organic Matter (DOM)

To further study the Fe_3_O_4_@CDA/O_3_ process, the 3D-EEM fluorescence spectra were employed to identify the organics of the biologically pre-treated landfill leachate. Figure 5 displayed the variations of the dissoluble organic matter in leachate before and after the process. Two strong EEM peaks (Ex/Em = 335/425 nm, Ex/Em = 250/450 nm) assigned to humic-like compounds (HA) and fulvic-like molecules (FA), respectively, were initially found [9,20]. The fluorescent peaks within the 3D-EEM spectra of the sample illustrated that the leachate before catalytic ozonation consisted of complex components which were related to the HA and the FA substances, which were the major part of the DOM [21,22,23,24]. The constituent of DOM in the leachate before catalytic ozonation showed that the influent still contained a lot of refractory compounds. As shown in Figure 2a, the fluorescence intensity of peak A was the highest, showing that the HA was the major contaminant in the leachate sample. After catalytic ozonation, two strong EEM peaks (Ex/Em = 335/425 nm, Ex/Em = 250/450 nm) almost vanished, illustrating that HA and FA were efficiently decomposed.

#### 3.2.2. Molecular Weight Variation of DOM in MLL in the Ozonation Process

For the sake of further characterization of the degradation and transformation patterns of refractory compounds in the leachate during the catalytic ozonation process, MW distribution was detected using HPLC (Figure 6). As shown in Figure 6, before catalytic ozonation, the DOM with the molecular weight region of >4 kDa in the leachate accounted for 90.7%, including HA and FA. MW > 10 kDa accounted for the remaining 10.30%. After catalytic ozonation, the DOM with MW > 10 kDa became completely decomposed, and the leachate was mainly composed of low MW substances (<4 kDa) such as volatile fat acids (VFAs) and protein-like substances, which accounted for 82.10%. These results suggested that macromolecular humic substances were easily attacked and degraded in the catalytic ozonation system, which led to the opening of aromatic rings and the formation of low molecular weight organic intermediates. However, some of the generated intermediates (mainly < 2 kDa) were not easily degraded. Obviously, organic compounds with high molecular weight were the major elements fighting against biodegradation [25]. Herein, the catalytic ozonation process played a significant role in improving the biodegradability of leachate through the efficient removal of these organic compounds.

### 3.3. Optimization of the MBR Process

#### 3.3.1. Treatment of Organics during the MBR Process

Hydraulic residence time (HRT) is the mean average residence time of the wastewater within the reactor, meaning the average reaction time of the wastewater with microbes in the bioreactor. The selection of HRT parameters is of great significance for the operation of the MBR process, and the optimal HRT helps to give full play to the advantages of the MBR process.

Figure 7 showed the variation of COD removal during the different HRT in the MBR reactor. As shown in Figure 7, only around 10% of COD removal was maintained at an HRT of 10 h. COD removal increased as the increased HRT, mainly because longer HRT partially reduced the organic pollution load within the reactor, extended the microbial and organic pollutant contact time, reduced the viscosity of activated sludge in the reactor, accelerated the oxygen and organic pollutant mass transfer rate, thus increasing COD removal. When the HRT is 28 h, the COD removal can reach around 85%.

#### 3.3.2. The Membrane Fouling

Hydraulic residence time (HRT) is also one of the important factors which can affect the contamination of membrane components. Extracellular polymer (EPS) has been identified as an important factor in membrane fouling in MBRs [26]. As shown in Figure 8, the EPS content in the MBR reactor gradually decreased with prolonged HRT. This phenomenon indicated that the longer HRT kept the organic load of the MBR reactor at low levels, which will help microbes reduce EPS generation and accelerate the degradation of the generated EPS. This result was partly beneficial to alleviate membrane pollution. At the same time, excessive proliferation of filamentous bacteria in activated sludge will cause increased EPS production, which will have serious adverse effects on the filtration of membrane components. However, during the trial process of this subsection, no signs of filamentous bacteria over-proliferation were found in the activated sludge by microscopy and SEM profiling within the MBR reactor. The activated sludge mixture in the MBR reactor was stored for 30 min for precipitation. The interface layer of activated sludge precipitation was clear, showing a good precipitation property. This also reflected that, even under longer HRT conditions, certain contaminants in leachate could still inhibit the over-proliferation of filamentous bacteria, which was partly conducive to alleviating the phenomenon of membrane pollution.

Consistent with previous studies [7,8], the HRT had an important influence on the membrane fouling rate. A recently reported study proved that changing the Solid retention time (SRT) and HRT could directly affect the rate of membrane fouling [27]. As was known to all, the parameters such as the morphology of the activated sludge, EPS, MLSS, and sludge viscosity had a major influence on the membrane fouling. For example, Campagna et al. [28] discovered that progressive reduction in HRT from 10–12 h to 3–4 h led to an obvious decrease in the dissolved oxygen concentration, which indirectly led to the result of excessive growth of filamentous bacteria in the activated sludge. This result, in turn, led to an increase in both EPS concentration and mixed liquor viscosity, which also led to a significant increment in membrane fouling at the low values of HRT. Wang et al. [26] found that abatement in HRT led to an increment in EPS and mean floc size, which indirectly led to a worsening of sludge settle ability and membrane resistance.

As shown in Figure 8, the prolongation of HRT has a significant impact on the activity and viscosity of activated sludge in the reactor. As HRT increases, the sour of activated sludge in the reactor gradually increases from 3.2 mg O_2_/g MLSS·h to 5.6 mg O_2_/g MLSS·h, which partly reflects the increased metabolic level of activated sludge utilization under larger HRT conditions. Under different HRT conditions, the SOUR variation is mainly closely related to the DO concentration within the MBR reactor and the properties of the activated sludge, including the sludge concentration and dynamic viscosity. The increment in the concentration and dynamic viscosity of activated sludge could affect the dissolution and diffusion between oxygen and contaminants in the mixture, thus causing adverse effects on the metabolic capacity of microorganisms [6].

As shown in Figure 8 and Figure 9, we found that both the viscosity and MLSS of the activated sludge within the MBR reactor decreased as HRT increased. In general, the shorter HRT, the more rapid proliferation of the sludge concentration in the reactor. In addition, extending the HRT of the reactor was conducive to controlling the viscosity of the activated sludge, ensuring the transmission of substances, and maintaining high activated sludge activity within the reactor. Previous research found that when HRT was extended to over 12 h, the concentration of activated sludge in the MBR reactor could be about 15,000 mg/L, which was consistent with the test results in this study [8].

As shown in Figure 10, the absorption peaks of the infrared map were concentrated in the range of 3100–2850 cm^−1^ (zone I) and 1600–900 cm^−1^ (zone Ⅱ). Specifically, the broad peak at 2950–2850 cm^−1^ was associated with the aliphatic C-H, C-H2, and C-H3 [29]. Meanwhile, the peak at 3100–3000 cm^−1^ was assigned to the presence of the aromatic C-H vibration. The results showed that the aromatic hydrocarbons in the leachate could be adsorbed by the activated sludge, the surface sludge of the membrane, and the filter cake layer sludge in this process. It has been shown that the absorption peak in the range of 1250–900 cm^−1^ is mainly associated with C-O-C, C-O vibration of polysaccharide, while the N-O vibration of nitrate in sludge mainly occurred at 1384 cm^−1^. C=C bonds, C=O bonds of the benzene ring structure, and C=O bonds of aldehydes, ketones, and esters could be related in the range of 1740–1640 cm^−1^.

As shown in Figure 11a, there was still an obvious pore structure on the membrane surface within the MBR reactor with catalytic ozonation after a period of operation. However, severe membrane hole blockage appeared on the surface of the membrane within the MBR reactor without catalytic ozonation. After catalytic ozonation, the DOM in the effluent varied greatly, which was partly conducive to alleviating the phenomenon of membrane pollution.

#### 3.3.3. Analysis of Bacterial Community Composition during the MBR Process

The metagenomic approach was employed to identify the taxonomic diversity of the activated sludge within the MBR reactor, which was used to seed the MBR system and was taken before and at the end of the experiment.

Before the experiment, the vaccinated sludge sample contained various microorganisms, such as organic matter degradation bacteria, ammonia-oxidizing bacteria, and nitrification bacteria. The results of the 16S rRNA gene sequencing showed that *Proteobacteria* (33.24%), *Proteobacteria* (26.71%), *Bacteroidetes* (9.36%), *Deinococcus-Thermus* (12.72%), *Actinobacteria* (4.67%), *Chloroflexi* (2.09%) were the dominant genera in the initial sludge (Figure 12). Similar observations were published by Zhao et al. [30], who found that *Proteobacteria*, *Proteobacteria*, and *Bacteroidetes* were the dominant genera in the anaerobic and anoxic pools when treating leachate.

Meanwhile, the 16S rRNA gene sequencing experiments confirmed that *Proteobacteria* (27.86%), *Proteobacteria* (34.67%), *Bacteroidetes* (10.59%), *Deinococcus-Thermus* (9.26%), *Actinobacteria* (5.24%), *Chloroflexi* (4.46%), *Gemmatimonadetes* (1.33%), *Nitrospirae* (5.98%), *Firmicutes* (1.26%) were the dominant genera in the MBR sludge at the end of the experiment under the condition of HRT of 28 h (Figure 8). The changes in the microbial community in both sludge samples could be clearly seen through the comparison, and the kinds of bacteria with sequences proportion more than 1% in the vaccinated sludge samples were consistent with the MBR membrane reactor sludge samples at HRT of 28 h. However, the proportion of *Gemmatimonadetes* and *Deinococcus-Thermus* decreased from 5.14% and 12.72% to 1.33% and 2.96%, respectively, which was consistent with previous research [29]. The most likely reason for this phenomenon was the change in the environment. In common, under the new environment or severe conditions, bacterial categories such as *Gemmatimonadetes* and *Deinococcus-Thermus* have a high proportion. However, under the reaction conditions with HRT of 28 h, the MBR reactor sludge was fully adapted, so the ratio of both decreased significantly. In addition, nitrite-oxidizing bacteria (NOB) were even less abundant in both initial and MBR reactor samples and were represented only by Nitrospira (0.49% and 5.98%, respectively), which can convert ammonia directly to nitrate (comammox process), as reported by Saleem et al. [31]. Nitration was guaranteed by the longer HRT and healthy operating conditions, contributing to the increase of the proportion of the *Nitrospirae*.

### 3.4. Removal of Contaminants by the Catalytic Ozonation-MBR System

There are many problems that existed in the original treatment process (a), such as membrane fouling, production of concentrated leachate, and the high cost of the membrane cleaning agent. In the present study, we try to investigate an effective and economical process to deal with landfill leachate. So, the catalytic micro-ozonation and the membrane bio-reactor (MBR) were employed as a further treatment for the leachate to meet the Standard for pollution control on the landfill site of municipal solid waste in China (GB 8978-1996). As shown in Table 1, the COD and NH_3_-N content in the effluent after MBR stabilized in the range of 57.85–65.38 mg/L and 5.98–10.24 mg/L, respectively. Obviously, the modified process flow is more simplified, replacing the ultrafiltration and nanofiltration processes.

## 4. Conclusions

Catalytic ozonation-MBR processes were effectively applied to treat the mature landfill leachate. Catalytic ozonation was conducted to investigate the impact of catalyst dosage and ozone concentration on removal performance. Results demonstrated that the organic pollutants were removed efficiently during the catalytic ozonation. MBR reactor was employed to treat the effluent after the catalytic ozonation. Under the condition of HRT of 28 h, the final effluent meets the Standard for pollution control on the landfill site of municipal solid waste in China (GB 8978-1996). Importantly, the membrane fouling within the MBR reactor was discussed under different HRTs. Moreover, the 16S rRNA gene sequencing experiments found that similar dominant phyla were detected before and at the end of MBRs. *Proteobacteria*, *Bacteroidetes*, and *Deinococcus-Thermus* counted up to 60% of the oxic and anoxic mixed community. The above findings suggest that the Catalytic ozonation–MBR integrated process showed strong feasibility in dealing with the stabilized landfill leachate.

## Figures and Tables

**Figure 1 toxics-10-00505-f001:**
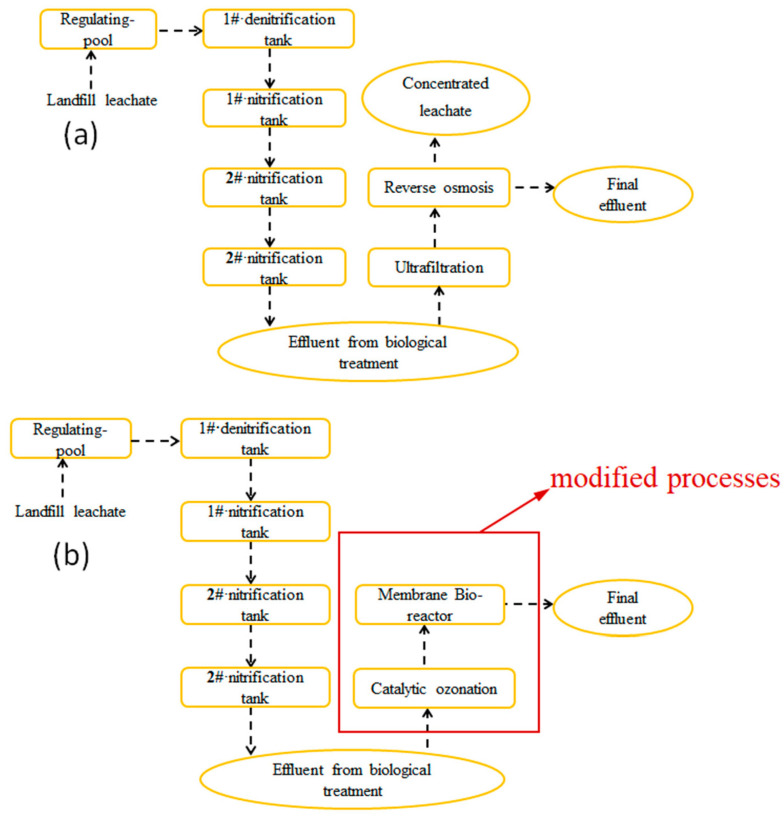
The leachate treatment process (**a**) the original procedure and (**b**) the used procedure of leachate in this study.

**Figure 2 toxics-10-00505-f002:**
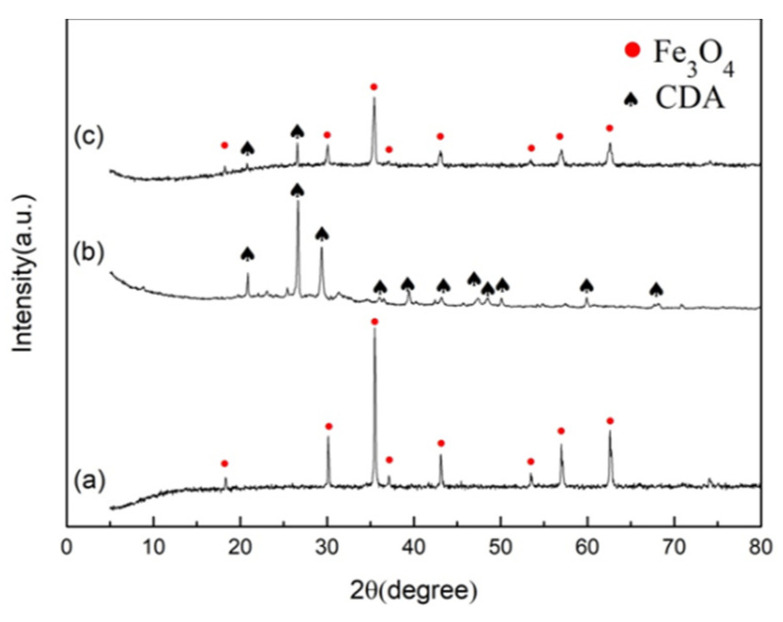
XRD patterns of (**a**) nano-Fe_3_O_4_, (**b**) CDA, (**c**) nano-Fe_3_O_4_@CDA.

**Figure 3 toxics-10-00505-f003:**
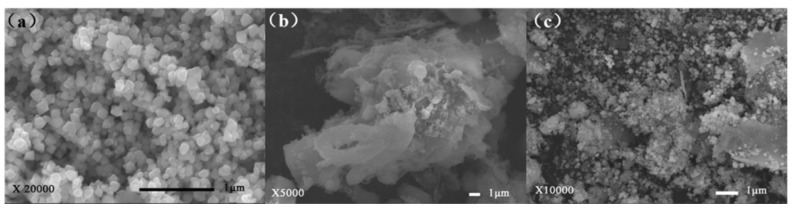
SEM of (**a**) nano-Fe_3_O_4_, (**b**) CDA, (**c**) nano-Fe_3_O_4_@CDA.

**Figure 4 toxics-10-00505-f004:**
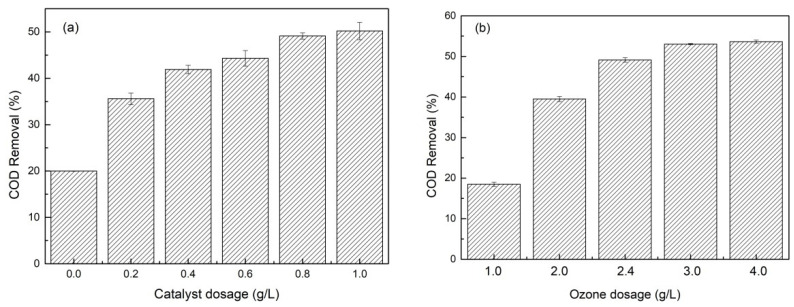
Effect of (**a**) catalyst dosage and (**b**) ozone dosage on COD removal of the biotreated leachate. Reaction condition (**a**): [O_3_] = 4.0 g/L; (**b**): [nano-Fe_3_O_4_@CDA] = 0.80 g/L. (Except for the investigated parameter, the other parameters were fixed at: [pH]_0_ = 8.4 ± 0.1, [reaction time] = 120 min).

**Figure 5 toxics-10-00505-f005:**
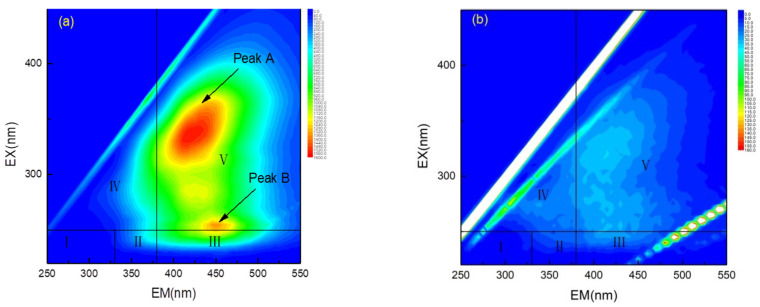
Three-dimensional excitation and emission matrix (3D-EEM) fluorescence spectra of biologically treated landfill leachate (**a**) before and (**b**) after catalytic ozonation (Reaction condition: [nano-Fe_3_O_4_@CDA] = 0.80 g/L, [O_3_] = 3.0 g/L, [pH] = 8.4 ± 0.1, [reaction time] = 120 min).

**Figure 6 toxics-10-00505-f006:**
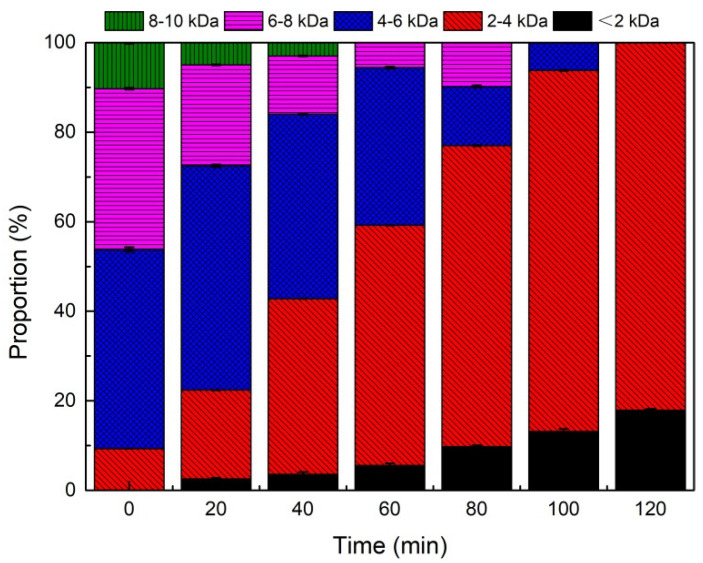
Molecular weight distribution of biologically pre-treated landfill leachate during catalytic ozonation. (Reaction condition: [nano-Fe_3_O_4_@CDA] = 0.80 g/L, [O_3_] = 3.0 g/L, [pH] = 8.4 ± 0.1, [reaction time] = 120 min).

**Figure 7 toxics-10-00505-f007:**
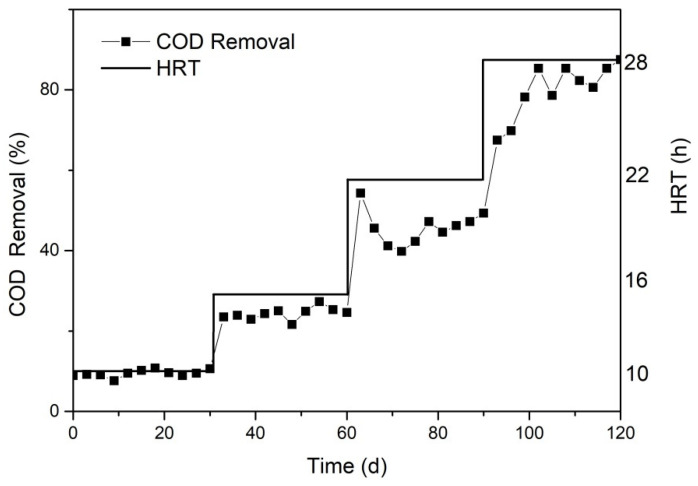
Effect of hydraulic residence time on COD removal by MBR process after catalytic ozonation. (Reaction condition: except for the investigated parameter, the other parameters were fixed at: [MLSS] = 10,000 mg/L, [DO] = 2.0 mg/L, [pH] = 8.4 ± 0.1).

**Figure 8 toxics-10-00505-f008:**
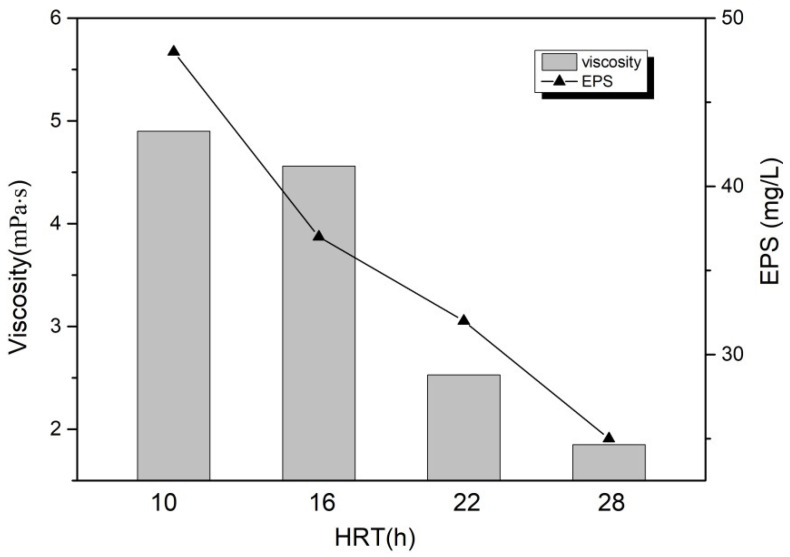
Effect of hydraulic residence time on viscosity and EPS by MBR process after catalytic ozonation. (Reaction condition: except for the investigated parameter, the other parameters were fixed at: [MLSS] = 10,000 mg/L, [DO] = 2.0 mg/L, [pH] = 8.4 ± 0.1).

**Figure 9 toxics-10-00505-f009:**
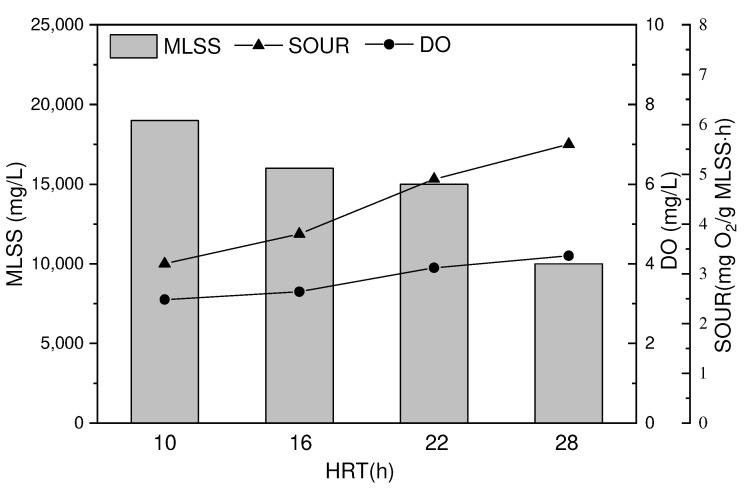
Effect of hydraulic residence time on MLSS, SOUR and DO by MBR process after catalytic ozonation. (Reaction condition: except for the investigated parameter, [pH] = 8.4 ± 0.1).

**Figure 10 toxics-10-00505-f010:**
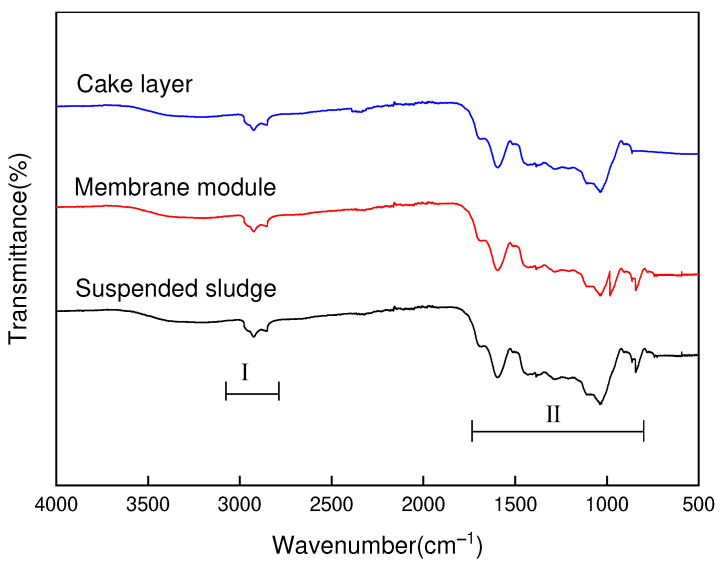
FTIR of activated sludge and membrane contaminants within MBR reactor.

**Figure 11 toxics-10-00505-f011:**
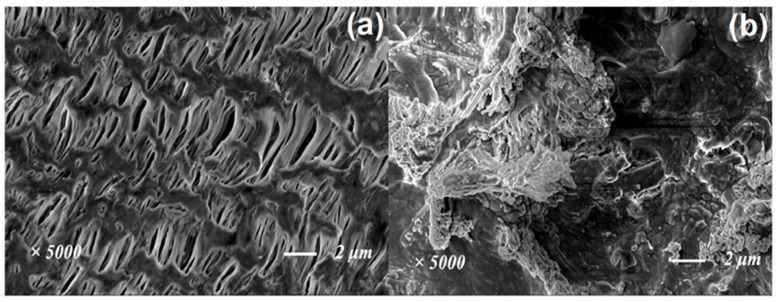
SEM of the membrane within MBR reactor with (**a**) and without (**b**) catalytic ozonation.

**Figure 12 toxics-10-00505-f012:**
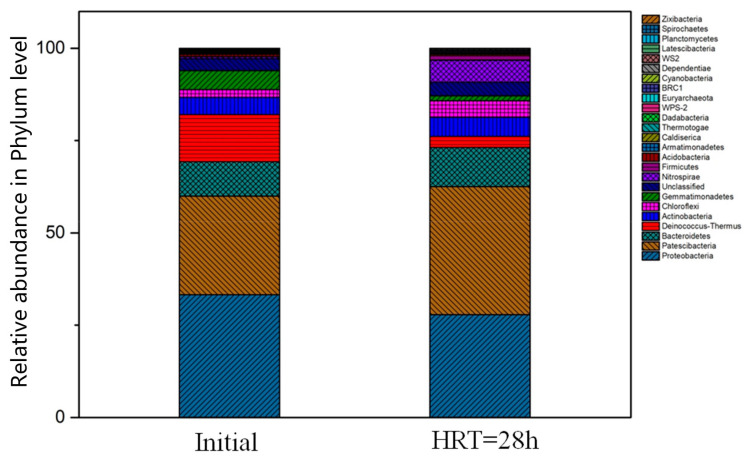
Relative abundance of phylum. (Reaction condition: [HRT] = 28 h, [pH] = 8.4 ± 0.1).

**Table 1 toxics-10-00505-t001:** Performance of different membrane processes in treating effluent.

Parameter	COD (mg/L)	NH_3_-N (mg/L)	Chromaticity (Times)	Cd (mg/L)	Cr (mg/L)	Pb (mg/L)
Std ^c^	100	25	40	0.01	0.1	0.1
MBR ^b^	62.64	8.97	6	ND	ND	ND
RO ^a^	25.35	2.75	3	ND	ND	ND

ND: non-detectable. ^a^: The original treatment process at the Houzhai landfill. ^b^: The modified procedure of leachate adopted by our group. ^c^: Standard for pollution control on the landfill site of municipal solid waste, China, GB 8978-1996.

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
