# Peer review of "A Combined Catalytic Ozonation-MBR Approach to Remove Contaminants from the Mature Landfill Leachate in the Yellow River Basin"

_toxics, 2022, doi:10.3390/toxics10090505_

Round 1

Reviewer 1 Report

The manuscript deals with catalyst ozonation in landfill leachate treatment.

Correct some characters (^ and @) that appeared improperly throughout the text

The catalyst was not described in the Methodology item.

MW distribution was detected using HPLC was not described in the Methodology item.

Figure 1 should be presented before appearing in the text

Item 3.1.3

When the humic grouping is attacked, the color of the leachate is visibly changed. Did the authors notice this?

Author Response

Correct some characters (^ and @) that appeared improperly throughout the text

As reviewer commented that we have carefully checked the manuscript and corrected some characters that improperly throughout the text in the change-tracked version.

The catalyst was not described in the Methodology item.

As reviewer commented that the details on the catalyst should be described clearly. We have carefully checked the manuscript and supplemented the concrete information about the catalyst in the part of “2.2. Preparation of the catalyst” and “2.5. Analytical methods” in the change-tracked version.

MW distribution was detected using HPLC was not described in the Methodology item.

As reviewer commented that we have carefully checked the manuscript and supplemented the concrete information about the MW distribution test in the part of “2.5. Analytical methods” in the change-tracked version.

Figure 1 should be presented before appearing in the text

As reviewer commented that we have adjusted the position of Figure 1 in the revised manuscript.

When the humic grouping is attacked, the color of the leachate is visibly changed. Did the authors notice this?

As reviewer commented that the color of the leachate was visibly changed when the humic grouping was attacked. We noticed this phenomenon during the catalyzation process.

Reviewer 2 Report

Minor revisions are needed for this manuscript to be published in the Toxics:

1) In the title please refer to the applied method of contaminants removal.

2) The landfill leachate is characterized by high concentrations of toxic organics, nitrogen oxides, and inorganic substances(Liu et al. 2018) – page 1 line 33. I would propose to list in brackets 2-3 examples of toxic organics.

3) Page 1, line 34 – oxides are underlined.

4) 2. Experimental section, 2.1. Leachate samples

“A sampling of the leachate was carried out during summer and autumn in 2020.” - Why samples were taken in summer and autumn? Do authors wanted to test the samples under different conditions (like e.g. temperature, humidity)? Whether it simply does not matter?

5) In the manuscript sometimes are spaces between a word and the name of the cited author, and sometimes there are no spaces. Please standardise this.

6) Figure 9 on page 10 is not clear. Please improve its quality.

Author Response

1) In the title please refer to the applied method of contaminants removal.

As reviewer commented that we have adjusted the title according to the applied method of contaminants removal in the change-tracked version.

2) The landfill leachate is characterized by high concentrations of toxic organics, nitrogen oxides, and inorganic substances(Liu et al. 2018) – page 1 line 33. I would propose to list in brackets 2-3 examples of toxic organics.

Many thanks to reviewer for the helpful suggestion, and we have carefully checked the manuscript and listed in brackets 2-3 examples of toxic organics in the change-tracked version.

3) Page 1, line 34 – oxides are underlined.

We have corrected it in the change-tracked version.

4) 2. Experimental section, 2.1. Leachate samples

“A sampling of the leachate was carried out during summer and autumn in 2020.” - Why samples were taken in summer and autumn? Do authors wanted to test the samples under different conditions (like e.g. temperature, humidity)? Whether it simply does not matter?

These experiments lasted nearly six months and were conducted during summer and autumn in 2020. We sampled during this period and it did not related to test the samples under different conditions.

5) In the manuscript sometimes are spaces between a word and the name of the cited author, and sometimes there are no spaces. Please standardise this.

We are sorry that the spaces in the manuscript were not standardized. We have carefully checked the manuscript and standardised it in the change-tracked version.

6) Figure 9 on page 10 is not clear. Please improve its quality.

As reviewer commented that Figure 9 on page 10 was not clearly. We have made some adjustments in the revised manuscript.

Reviewer 3 Report

Recommendation: The manuscript may be publishable after a major revision

Cui Ma and et al. report on optimization of MBK process of leachate samples from a landfill in Yuanyang County in Xinxiang. This article is prepared for a special issue devoted to Pollution Control in the Yellow River Basin. Environmental problems in China of special importance because this country with dense population is intensively developing industry including chemiclas and producing many substances which is now distributed around the world. Thus, a strict control on pollution in China is particular important and contributions of chinses scholars who are familiar with local environmental conditions are welcome and of great interest. The results of biological treatment of landfill leachate are dependent on its composition and can be inefficient under certain conditions. Therefore, other cheap strategies should be developed to complete available approaches to a decontamination of wasters. Insertion of oxidation steps in MBK process is attracting great attention to improve the decomposition of organic compounds. The authors investigated different conditions of catalytic ozonolysis of leachate samples from a landfill in Yuanyang County in Xinxiang to optimized MBK degradation process. The work is well organized and the article simple to read but many experimental details are missing. Thus, the experiments cannot be repeated. This work can be published in Toxics only after extensive editing of the manuscript. Missing information should be added. I recommend this manuscript to publication only after corrections of following points

1.     Please, precise the nature of the Fe3O4@CDA catalyst (composition, porosity, method of preparation or reference on the preparation). Without this information the article does not have any scientific meaning and cannot be published as a scientific article.

2.     Figure 1. Please, precise pH, time and other experimental conditions used in these experiments in the capture (the figures should be readable without the main text).  Please, precise exact conditions in which catalytic activity was investigated (rate of production of O3, for example) and ozone dosage was investigated (amount of the catalyst). Without this information the experiments cannot be reproduced.

3.     Figures 2-6 and 8. Please precise in the caption of these Figures the experimental conditions in which the sample under studies was obtained.

4.     Section 3.2.1. Treatment of organics during the MBR process. Please, describe the experimental conditions (pH, residual time in each reactor, detailed conditions of ozonolyse and MKB process.

5.     Please, compare the membrane fouling and other characteristics with and without catalytic ozonation.

There are also minor points to correct:

1.     Please, reformate references using MDPI format in the text.

2.     line 8. Please, complete the e-mails of corresponding authors.

3.     line 20 and 45. Please, explain the meaning of COD abbreviation (it appears only in line 110 but should be given here).

4.     line 53. Please, explain the meaning of NH3-N it appears only in line 110 but should be given here) and chroma.

5.     line 78-80. Please reformulate the sentence. If the aim is a comparison, at least two samples should be discussed in this statement. It seems that the goal is studies of different conditions for the decomposition of secondary biological effluent of landfill leachate.

6.     line 90 : was shown in Figure 9. Two processes are shown in this figure. Please, give an explanation which was used in this work. Change the numbering of Figures. This is the first figure which is mentioned in the text. Thus, it should have the number 1 and should be move in this part of the article. Figure 9 describe the whole process which was not investigated in this work. The comment should be added in the text.

7.     line 91. Please, explain the meaning of BOD abbreviation.

8.     line 142. Please, add a space after “Ma et al. 2019).

9.      line 149 “The cause was most likely that a portion  of •OH was consumed by the rest of O3.” Please, give the reference or corresponding equations or delete this sentence.

10.  line 143 “In addition, the active sites of Fe3O4@CDA were common to be limited under larger O3 dosage conditions (Ma et al. 2020).” Please, precise your explaination which is unclear in the present form.

11.  line 152. Please, explain the meaning of DOM abbreviation.

12.  line 177 “such as  VFAs and protein-like substances”. Please explain the meaning of VFA abbreviation. How the nature of these compounds was determined?

13.  line 209-213. Please, separate this sentence in several phrases (reformulate the text). The sense is unclear.

14.  line 219. Please, add a space after 30 (30 min).

15.  line 226. Please, explain the meaning of SRT abbreviation.

Author Response

1.  Please, precise the nature of the Fe3O4@CDA catalyst (composition, porosity, method of preparation or reference on the preparation). Without this information the article does not have any scientific meaning and cannot be published as a scientific article.

As reviewer commented that the details on the catalyst should be described clearly. We have carefully checked the manuscript and supplemented the concrete information about the catalyst in the part of “2.2. Preparation of the catalyst” and “3.1. Characterization of the catalyst” in the change-tracked version.

  1. Figure 1. Please, precise pH, time and other experimental conditions used in these experiments in the capture (the figures should be readable without the main text).  Please, precise exact conditions in which catalytic activity was investigated (rate of production of O3, for example) and ozone dosage was investigated (amount of the catalyst). Without this information the experiments cannot be reproduced.

We are sorry that the experimental conditions on the pollutant degradation were not exhibited in the capture and in the change-tracked version the exact conditions on the pollutant degradation has been supplemented in the change-tracked version..

  1. Figures 2-6 and 8. Please precise in the caption of these Figures the experimental conditions in which the sample under studies was obtained.

As reviewer commented that we have precised in the caption of these Figures the experimental conditions in which the sample under studies was obtained in the change-tracked version.

  1. Section 3.2.1. Treatment of organics during the MBR process. Please, describe the experimental conditions (pH, residual time in each reactor, detailed conditions of ozonolyse and MKB process.

As reviewer commented that we have described the experimental conditions (pH, residual time in each reactor, detailed conditions of ozonolyse and MBR process.

  1. Please, compare the membrane fouling and other characteristics with and without catalytic ozonation.

As reviewer commented that we have supplemented the comparison about the SEM of the membrane by MBR process with and without catalytic ozonation in Figure 11. The SEM of the membrane showed the membrane surface morphology clearly, which pointed up the difference between with and without catalytic ozonation. Related the experimental details and results of the membrane fouling and other characteristics with and without catalytic ozonation have been investigated in the further research by our research group. We cordially invite the reviewer to review our further research work about the special research on the comparison of the membrane fouling and other characteristics with and without catalytic ozonation and many thanks to the reviewer.

There are also minor points to correct:

  1. Please, reformate references using MDPI format in the text.

Many thanks to reviewer for the helpful suggestion, and we have reformated references using MDPI format in the change-tracked version.

  1. line 8. Please, complete the e-mails of corresponding authors.

We have completed the e-mails of corresponding authors in the change-tracked version.

  1. line 20 and 45. Please, explain the meaning of COD abbreviation (it appears only in line 110 but should be given here).

Many thanks to reviewer for the helpful suggestion, and we have explained the meaning of COD abbreviation in the change-tracked version.

  1. line 53. Please, explain the meaning of NH3-N it appears only in line 110 but should be given here) and chroma.

Many thanks to reviewer for the helpful suggestion, and we have explained the meaning of NH3-N in the change-tracked version.

  1. line 78-80. Please reformulate the sentence. If the aim is a comparison, at least two samples should be discussed in this statement. It seems that the goal is studies of different conditions for the decomposition of secondary biological effluent of landfill leachate.

Many thanks to reviewer for the helpful suggestion, and we have reformulatde the sentence in the change-tracked version.

  1. line 90 : was shown in Figure 9. Two processes are shown in this figure. Please, give an explanation which was used in this work. Change the numbering of Figures. This is the first figure which is mentioned in the text. Thus, it should have the number 1 and should be move in this part of the article. Figure 9 describe the whole process which was not investigated in this work. The comment should be added in the text.

As reviewer commented that we have explained the two process which was used in this work and changed the numbering of Figures in the change-tracked version.

  1. line 91. Please, explain the meaning of BOD abbreviation.

Many thanks to reviewer for the helpful suggestion, and we have explained the meaning of BOD in the change-tracked version.

  1. line 142. Please, add a space after “Ma et al. 2019).

Many thanks to reviewer for the helpful suggestion, and we have added a space after “Ma et al. 2019) in the change-tracked version.

.

  1. line 149 “The cause was most likely that a portion  of •OH was consumed by the rest of O3.” Please, give the reference or corresponding equations or delete this sentence.

Many thanks to reviewer for the helpful suggestion, and we have deleted this sentence in the change-tracked version.

  1. line 143 “In addition, the active sites of Fe3O4@CDA were common to be limited under larger O3dosage conditions (Ma et al. 2020).” Please, precise your explaination which is unclear in the present form.

We have precise the related explanation the change-tracked version.

  1. line 152. Please, explain the meaning of DOM abbreviation.

Many thanks to reviewer for the helpful suggestion, and we have explained the meaning of DOM in the change-tracked version.

  1. line 177 “such as  VFAs and protein-like substances”. Please explain the meaning of VFA abbreviation. How the nature of these compounds was determined?

Many thanks to reviewer for the helpful suggestion, and we have explained the meaning of VFAs in the change-tracked version. The nature of these compounds could be determined by several methods such as gas phase chromatography, colorimetry, titrimetry.

  1. line 209-213. Please, separate this sentence in several phrases (reformulate the text). The sense is unclear.

As reviewer commented that we have reformulated the text in the change-tracked version.

  1. line 219. Please, add a space after 30 (30 min).

Many thanks to reviewer for the helpful suggestion, and we have added a space after 30 in the change-tracked version.

  1. line 226. Please, explain the meaning of SRT abbreviation.

Many thanks to reviewer for the helpful suggestion, and we have explained the meaning of SRT in the change-tracked version.

Round 2

Reviewer 3 Report

Recommendation: The manuscript can be published in the present form

Cui Ma and et al. carefully revisited the article. All my remarques were considered and the text is nicely corrected to be clear for readers. The article can be published in the present form.